# Treatment with Glyphosate Induces Tolerance of Citrus Pathogens to Glyphosate and Fungicides but Not to 1,8-Cineole

**DOI:** 10.3390/molecules27238300

**Published:** 2022-11-28

**Authors:** Nor EL Houda Tahiri, Hamza Saghrouchni, Noureddine Hamamouch, Mostafa El Khomsi, Abdulhakeem Alzahrani, Ahmad Mohammad Salamatullah, Lyoussi Badiaa, Lalla Aicha Lrhorfi

**Affiliations:** 1Laboratory of Biochemistry, Biotechnologies and Health and Environment, Department of Biology, Faculty of Sciences, Ibn Tofail University, B.P. 133, Kenitra 14000, Morocco; norelhouda.tahiri@uit.ac.ma (N.E.H.T.); alrhorfi_com@yahoo.fr (L.A.L.); 2Laboratory of Natural Substances, Pharmacology, Environment, Modeling, Health and Quality of Life, Faculty of Sciences Dhar El Mahraz, University of Sidi Mohamed Ben Abdellah, Fez 30000, Morocco; n.hamamouch@ulster.ac.uk (N.H.); lyoussi@gmail.com (L.B.); 3Department of Biotechnology, Institute of Natural and Applied Sciences, Çukurova University, Balcalı/Sarıçam, 01330 Adana, Turkey; 4Department of Entomology and Plant Pathology, North Carolina State University, Raleigh, NC 27695, USA; 5Natural Resources and Sustainable Development Laboratory, Department of Biology, Faculty of Sciences, Ibn Tofail University, B.P. 133, Kenitra 14000, Morocco; elkhomsi.mostafa@uit.ac.ma; 6Department of Food Science & Nutrition, College of Food and Agricultural Sciences, King Saud University, 11 P.O. Box 2460, Riyadh 11451, Saudi Arabia; aabdulhakeem@ksu.edu.sa (A.A.); asalamh@ksu.edu.sa (A.M.S.)

**Keywords:** *Penicillium italicum*, *Penicillium digitatum*, *Geotrichum candidum*, 1,8-cineole, fungicide tolerance, cross-resistance, postharvest

## Abstract

During the postharvest period, citrus fruits are exposed to *Penicillium italicum*, *Penicillium digitatum,* and *Geotrichum candidum*. Pesticides such as imazalil (IMZ), thiabendazole (TBZ), orthophenylphenol (OPP), and guazatine (GUA) are commonly used as antifungals. Glyphosate (GP) is also used in citrus fields to eliminate weed growth. The sensitivity of fungal pathogens of citrus fruit to these pesticides and 1,8-cineole was evaluated, and the effect of GP on the development of cross-resistance to other chemicals was monitored over a period of 3 weeks. IMZ most effectively inhibited the mycelial growth and spore germination of *P. digitatum* and *P. italicum*, with minimum inhibitory concentrations (MICs) of 0.01 and 0.05 mg/mL, respectively, followed by 1,8-cineole, GP, and TBZ. 1,8-Cineole and GP more effectively inhibited the mycelial growth and spore germination of *G. candidum*, with minimum inhibitory concentrations (MICs) of 0.2 and 1.0 mg/mL, respectively, than OPP or GUA. For the spore germination assay, all substances tested showed a total inhibitory effect. Subculturing the fungal strains in culture media containing increasing concentrations of GP induced fungal tolerance to GP as well as to the fungicides. In soil, experiments confirmed that GP induced the tolerance of *P. digitatum* to TBZ and GP and the tolerance of *P. italicum* to IMZ, TBZ, and GP. However, no tolerance was recorded against 1,8-cineole. In conclusion, it can be said that 1,8-cineole may be recommended as an alternative to conventional fungicides. In addition, these results indicate that caution should be taken when using GP in citrus fields.

## 1. Introduction

The global production of Citrus spp. reaches 143 million tonnes, and there are around 140 citrus-producing countries, among which the main producers are Brazil, the Mediterranean region, the USA, and China. These countries alone produce 60% of global production [1]. Among citrus species, oranges occupy the majority share of the world’s citrus production (55%), followed by mandarins (25%), lemons (13%), and grapefruits (7%) [2]. The main citrus-producing countries in the Mediterranean region include Spain, Egypt, Italy, Turkey, and Morocco [3]. Around 20 million tons of citrus and lime and 4.4 million tons of grapefruit and pomelos are produced in the world [1].

In Morocco, the citrus farming sector is considered one of the most important agricultural sectors. It makes a significant contribution to Gross Domestic Product (GDP) growth, which leads to national socioeconomic development [4,5]. It constitutes the main source of income for 10.000 families. The exportation of citrus between 2015 and 2019 is estimated at 644.000 tons and accounts for 44,96 $ million annually. The current area of production is estimated at 128.000 hectares, and the average annual production is around 2.4 million tons. The varietal profile of Moroccan citrus orchards is represented by three main varieties: clementine (35%), oranges (21%), and navel (18%) [4].

However, this diversity of citrus varieties, together with climate conditions, favours the development of a variety of pathogens, especially fungal pathogens, which are known to cause significant yield losses of important crops [6]. Fungal pathogens can infect plants, and they are capable of infecting crops in areas where humidity and temperature are high, such as Morocco, and can cause up to a 20% yield loss [7].

During storage and transport, citrus fruits are susceptible to blue, green, and soft rot diseases due to attacks by *Penicillium italicum* (*P. italicum*), *Penicillium digitatum* (*P. digitatum*)*,* and *Geotrichum candidum* (*G. candidum*). These diseases can spread during the postharvest period, producing several classes of mycotoxins as secondary metabolites [8,9,10].

To control these rots in the postharvest period, citrus growers and exporters rely mainly on chemical antifungals. The most frequently used chemical treatments are thiabendazole (TBZ), imazalil (IMZ), orthophenylphenol (OPP), and guazatine (GUA) [11,12,13]. However, these chemical treatments have many drawbacks, the most serious of which is the persistence of chemical residues in the fruits, which poses serious problems to human health and the environment [14,15]. In addition, the indiscriminate use of chemical fungicides on postharvest citrus fruit leads to the development of fungal strains resistant to commonly used fungicides, and this presents a serious challenge, making the chemical control system ineffective [16]. As a result, resistance has become an important factor limiting the efficacy and lifespan of fungicides [17,18].

Glyphosate (*N*-phosphonomethyl glycine), a small molecule, has become the dominant herbicide worldwide, especially since the introduction of transgenic glyphosate-resistant crops in the 1990s [19,20]. GB was patented by Monsanto as an antibiotic in 2010 (US Patent No. 7771736 B2) because it can be used against a wide spectrum of microorganisms and protozoa [21]. Glyphosate residues can spread widely and accumulate in the soil, water, and plant products, raising concerns over human and environmental health [22,23]. Although designed to control weed growth, GP also affects microorganisms that use the herbicide’s molecular target, the enzyme enolpyruvyl-shikimate-3-phosphate synthase, to synthesise aromatic amino acids [24]. Studies have shown increased resistance, co-resistance and cross-resistance to antibiotics and fungicides after exposure to high concentrations of GP and other herbicides [25,26,27,28,29,30]. Resistance to GP is conferred, in particular, by multidrug efflux pumps [31].

More seriously, the chemical treatment of citrus in the field, for example, with GP, can lead to the development of tolerance not only to GP but also to other fungicides used in postharvest treatments, a phenomenon referred to as cross-resistance. The study hypothesis is that the use of GP in the field contributes to the emergence of fungi resistant to other chemical antifungal agents used in the postharvest storage and packaging of citrus fruits in particular.

The above-mentioned problems make it necessary to look for ways to control postharvest diseases in citrus fruit. This requires research into new and more active molecules, such as essential oils, their main compounds, and plant extracts. Among others, 1,8-cineole (eucalyptol), a monoterpene cyclic ether, which was used in this study, is commonly found in many plants with essential oils, such as plants belonging to the genera *Rosmarinus*, *Eucalyptus*, *Cinnamomum,* and *Laurus* [32,33,34,35].

To deal with this problem, the objectives of this research are twofold: (1) to determine the sensitivity levels of *P. italicum, P. digitatum,* and *G. candidum* to glyphosate and commonly used fungicides and (2) to examine the effect of pre-treatment with glyphosate on fungal tolerance to the above-mentioned fungicides.

## 2. Results

### 2.1. Isolation and Identification of Fungi Responsible for Postharvest Citrus Fruit Rot

The identities of the fungal strains were determined morphologically at the macro and microscopic levels. The characteristic appearances of *Penicillium* and *Geotrichum* genera were identified.

Colonies of *P. italicum* possess a grey-blue, dense, powdery, and downy appearance. Colonies of *P. digitatum* have a grey-green, dense, powdery, and fluffy appearance, while those of *G. candidum* have a white, smooth, and thin appearance (Figure 1). The reverse side is white and has radiations from the centre. At the microscopic level, *Penicillium* has erect, more or less branched, and terminated conidiophores of phialides. The phialides are arranged in verticils at the tips of the conidiophores. They are attached to the same conidiophores, or the phialides are tightly packed together, forming a brush or pencil image. In *G. candidum*, the hyphae have joints and disarticulate by schizolysis, producing chains of arthroconidia. These arthroconidia are unicellular, smooth, walled, more or less cylindrical in shape, rounded at the extremities, and of variable sizes.

### 2.2. Antifungal Activity

Imazalil exhibited strong antifungal activity against *P. digitatum* and *P. italicum*, with MICs of 0.01 mg/mL and 0.05 mg/mL, respectively, for the two species, indicating high sensitivity levels of *Penicillium* strains to IMZ (Table 1). However, for TBZ, MICs of 1.5 mg/mL and 0.375 mg/mL were necessary to achieve the complete inhibition of *P. digitatum* and *P. italicum*, respectively (Table 1). A 75% inhibition of *P. digitatum* was observed when TBZ was used at 1.75 mg/mL, and less than 50% inhibition was obtained when lower concentrations were used (Figure 2B). These results indicate that *P. italicum* and *P. digitatum* are very sensitive to IMZ compared to TBZ.

The minimal inhibitory concentration of GUA to completely inhibit the growth of *G. candidum* was 4 mg/mL (Table 1). Between 50% and 75% inhibition was observed when concentrations between 2.25 and 3 mg/mL were used (Figure 2C).

Orthophenylphenol exhibited stronger antifungal activity against *G. candidum* compared to GUA. A MIC of 1.5 mg/mL was sufficient to completely inhibit *G. candidum* growth (Table 1). Less than 75% inhibition was observed when the concentration of OPP used was 0.75 mg/mL, and less than 50% inhibition was observed when the concentration of OPP was below 0.375 mg/mL (Figure 2D). These results indicate that OPP is very effective at low concentrations against *G. candidum* compared to GUA.

1,8-Cineole with MIC values between 0.2 and 0.4 mg/mL caused the complete inhibition of all fungal pathogens (Table 1). The results in Figure 2E show that the percentage of inhibition (PI) increases with increasing concentration. In addition, the figure reflects a partial inhibition of almost 50% with relatively low concentrations of 0.1 and 0.2 mg/mL against the fungal strains.

Surprisingly, GP appeared to be very effective against all of the fungal pathogens tested. The observed MIC of GP against both strains of *Penicillium* was 0.24 mg/mL, which is higher than the MIC value of IMZ but lower than the MIC of TBZ (Table 1). A 70% inhibition of *P. italicum* was observed when GP was used at 0.12 mg/mL (Figure 2F). This result indicates that IMZ is very efficient against *P. digitatum* and *P. italicum*, followed by GP, 1,8-cineole, and then TBZ.

GP was also very effective against *G. candidum* with a MIC of 1 mg/mL, which is lower than the MIC values of both OPP and GUA, indicating that GP and 1,8-cineole are more effective than both GUA and OPP for the control of *G. candidum*. The MIC values of the tested antifungal agents are summarised in Table 1.

### 2.3. Inhibition of Spore Germination

The effects of IMZ, TBZ, GUA, OPP, 1,8-cineole, and GP on the spore germination of the different fungal pathogens were also examined. Different concentrations, which were very close to or equal to MICs, were tested, and the number of colonies was counted as a measure of spore germination inhibition.

The complete inhibition of P. digitatum and P. italicum spore germination was achieved when IMZ was used at 0.1 mg/mL, followed by 1.8-cineole and GP at 0.4 and 0.5 mg/mL, respectively (Table 2 and Table 3), confirming the efficacy of IMZ and GP against both P. digitatum and P. italicum. Higher concentrations of TBZ, in the order of 1.5 mg/mL and 0.75 mg/mL, were needed to completely inhibit the spore germination of P. digitatum and P. italicum, respectively (Table 2 and Table 3).

1,8-Cineole completely inhibited the spore germination of *G. candidum* at a concentration of 0.2 mg/mL, followed by GP at 1 mg/mL, OPP at 1.5 mg/mL, and then GUA at 4 mg/mL, confirming that GP and OPP are very effective against *G. candidum* compared to GUA (Table 4).

### 2.4. The effect of Glyphosate on Fungal Sensitivity to Different Antifungals In Vitro

The ability of *P. digitatum*, *P. italicum*, and *G. candidum* to develop tolerance to GP was examined in vitro by subculturing the fungal strains in a culture medium containing increasing concentrations of GP. All fungal strains seemed to develop tolerance to GP after subculturing. The MIC of GP doubled from 0.24 mg/mL before subculturing to 0.5 mg/mL after subculturing for *P. italicum* and from 1 mg/mL to 2 mg/mL for *G. candidum*. However, the MIC of GP against *P. digitatum* quadrupled after subculturing, increasing from 0.24 mg/mL to 1 mg/mL, indicating the ability of *P. digitatum* to quickly develop tolerance to GP compared to the other fungal strains (Table 5).

The fact that the fungal strains were able to develop tolerance to GP prompted us to test whether they can develop tolerance to the other antifungals.

The results indicated that *P. digitatum* and *P. italicum* developed tolerance to IMZ, as indicated by the MIC values before and after subculturing; the MIC of IMZ to completely inhibit *P. digitatum* and *P. italicum* increased from 0.01 to 0.4 mg/mL for the former and from 0.05 to 0.8 mg/mL for the latter. *P. italicum* also developed tolerance to TBZ, as indicated by the increase in the MIC value from 0.375 to 3 mg/mL (10 times). However, *P. digitatum* did not show any tolerance towards TBZ (Table 6). *G. candidum* also developed tolerance to GUA and OPP. After subculturing, the MIC of GUA and OPP to completely inhibit *G. candidum* doubled, increasing from 1.5 to 3 mg/mL for GUA and from 1 to 2 mg/mL for OPP (Table 6). Taken together, these results indicate that GP can induce the tolerance of *P. digitatum*, *P. italicum*, and *G. candidum* to the used fungicides in vitro. On the other hand, no change in MIC was detected for 1,8-cineole.

### 2.5. The Effect of Glyphosate on Fungal Sensitivity to Different Antifungals In Vitro in Contaminated Soil

To examine the effect of GP soil application on the tolerance level of Penicillium to the fungicides, three different concentrations of GP were applied to the soil: 0.12 mg/mL, 0.24 mg/mL, and 0.5 mg/mL, and the MICs of IMZ, TBZ, and 1,8-cineole were calculated every week for a period of three weeks.

At 0.12 mg/mL GP, the MIC values of IMZ and TBZ antifungal agents did not change, even after three weeks of GP application (Figure 3A,B). However, when the GP concentration increased to 0.24 mg/mL, after 3 weeks of GP soil application, *P. digitatum* started to develop tolerance to GP, as indicated by the MIC value, which increased from 0.5 mg/mL in week 1 to 1.5 mg/mL in weeks 2 and 3. *P. digitatum* also started to develop tolerance to TBZ two weeks after GP application (Figure 3C).

At a higher concentration of GP (0.5 mg/mL), *P. digitatum* started to develop tolerance to TBZ immediately after GP application, increasing the MIC from 3 mg/mL in week 1 to 4 mg/mL in weeks 2 and 3. However, *P. digitatum* was still sensitive to TBZ and GP, even after several days of GP soil application (Figure 3D). For 1,8-cineole, no tolerance to the antifungals developed in *P. digitatum* (Figure 3).

Soil treatment with 0.12 mg/mL GP did not induce the tolerance of *P. italicum* to the antifungal agents when compared to the control (Figure 4A,B). However, when GP was present in the soil at 0.24 mg/mL, the MIC value for IMZ against *P. italicum* increased sharply from 0.05 mg/mL to 0.65 mg/mL in the first week and remained stable after that (Figure 4C). The MIC value for GP also increased from 0.25 mg/mL in the first week following soil treatment with GP to 0.65 mg/mL by the third week. This result indicates that *P. italicum* can develop tolerance to IMZ and GP when soil is treated with 0.24 mg/mL GP. *P. italicum* did not show tolerance to TBZ (Figure 4C).

When the soil was treated with a higher concentration of GP (0.5 mg/mL), *P. italicum* developed tolerance to all three antifungal agents, GP, IMZ, and TBZ. The MICs of these antifungal agents increased from values below 0.5 mg/mL to 1 mg/mL for GP and TBZ and to 1.5 mg/mL for IMZ (Figure 4D). For 1.8-cineole, no tolerance to the antifungals developed in *P. italicum* (Figure 4).

## 3. Discussion

The citrus sector in Morocco has been growing rapidly in recent years thanks to the agricultural development strategy supported by the Green Morocco Plan [4]. This strategy consists of increasing the area cultivated with citrus fruits in Morocco. However, citrus cultivation is prone to challenges commonly posed by diseases in the pre- and postharvest periods, which has a detrimental effect on the quantity and quality of the harvested fruit. In recent years, several studies have focused on screening plant extracts with the objective of developing novel antifungal compounds that can be used in the control of citrus postharvest diseases [36].

To combat plant diseases caused by fungal agents, various fungicides are used. However, these fungicides are now either prohibited or are in the process of being prohibited because of the risk of their toxicity to human consumers and the environment and due to the emergence of microorganisms that are increasingly becoming resistant to the authorised doses of these fungicides. In addition, the use of GP in fields where citrus trees are grown can pose additional problems, such as the tolerance of fungal pathogens to postharvest fungicides. It is in this context that we have opted to test GP, 1,8-cineole, and different fungicides on citrus fruit fungi and to test the effect of GP application on fungal tolerance to the fungicides.

The evaluation of the antifungal activity showed that all three strains were sensitive to relatively low concentrations of postharvest antifungals. IMZ was very effective on *P. digitatum* and *P. italicum*, with MICs of 0.01 and 0.05 mg/mL, respectively, followed by 1,8-cineole, GP, and then TBZ. A previous study conducted by [37] on the effect of *Thymus* essential oil against *P. digitatum* and *P. italicum* found a MIC value of 0.13 mg/mL. Essential oils of *Origanum syriacum* and *Foeniculum vulgare* have been shown to inhibit the conidial germination and germ tube elongation of *P. digitatum* at a concentration of 64 μg/mL. Similarly, *Origanum* and *Foeniculum* oils, at concentrations of 64 and 352 μg/mL, were found to completely inhibit germ tube elongation. Microscopic observations revealed that *Oreganum* and *Foeniculum* oils significantly altered the morphology of the hyphae of *P. digitatum*. [38] reported the total inhibition of the spore germination of *P. digitatum*, *P. italicum*, and *G. candidum* after the exposure of the spores to 6 mL/L and 15 mL/L citral solutions for 1 h.

On the other hand, 1,8-cineole and GP proved to be very effective against *G. candidum*, followed by OPP and GUA. It has been reported that the aqueous extracts of *Cistus villosus* and *Halimium antiatlanticum* are effective against *G. candidum*, with a MIC of 0.156 mg/mL [5]. The study by [36] worked on *Geotrichum citri-aurantii*. Indeed, the PI results were 22% for 1,8-cineole, 42% for TBZ, 48% for *Rosmarinus officinalis*, 41% for *Eucalyptus radiata*, and 35% for *Cinnamomum camphora*. All of these plants contain a significant quantity of 1,8-cineole. In the same study, four essential oils were used to evaluate the effect on spore germination, and the tested oils completely inhibited the germination of *G. citri-aurantii* spores per 1 mL/L. However, two other oils resulted in total inhibition at concentrations of 0.6 and 0.5 mL/L. In the study of [39], fungal sporulation was reduced to 22.5% and 25% for *P. italicum* and *P. digitatum*, respectively, at 50 mg/mL of neroli oil.

The in vitro subculturing of the fungal strains in culture media containing increasing concentrations of GP induced fungal tolerance to GP as well as to the fungicides, except for TBZ in the case of *P. digitatum.* In soil, experiments confirmed that GP can induce the tolerance of *P. digitatum* to TBZ and GP, the tolerance of *P. italicum* to IMZ, TBZ, and GP, especially when it is present in the soil at 0.5 mg/mL, and the tolerance of *G. candidum* to OPP and GUA. However, no increase in MIC with respect to 1,8-cineole was observed. Such a phenomenon of cross-resistance is an aggravating factor. These results confirmed the hypothesis that the use of GP in the field contributes to the emergence of fungi resistant to other chemical antifungal agents used in the postharvest storage and packaging of citrus fruits in particular.

Several studies have been carried out in order to understand the mechanisms of resistance development. In the study of [40], the evaluation of the sensitivity of 75 strains of *P. digitatum* to seven different fungicides, Azoxystrobin, Fludioxonil, Imazalil, Myclobutanil, Prochloraz, Thiabendazol, and Trifloxystrobin, showed a significant number of strains resistant to TBZ (84%), IMZ (77%), and the most common fungicides used during the citrus fruit postharvest period. The molecular characterisation of different *P. digitatum* genes involved in fungicide resistance were carried out. All P. digitatum genes were selected on the basis of particular resistance mechanisms due to the target or mode of action of the fungicide. TBZ resistance was characterised by a single point mutation in the gene sequence of β-tubulin corresponding to amino acid 200, confirming previous work on this subject. In all cases, the resistance mechanism was consistent in the isolates from the orchard or packing station, and no differences conferred by the fungicide origin or pressure were observed.

Other in-depth molecular studies have been carried out to explain the mechanisms of fungicide tolerance and to develop efficient and rapid methods for the detection of resistant genotypes in fungal pathogens [41,42]. Moreover, researchers have attempted to explain fungicidal mechanisms in order to avoid an increase in resistant populations [43,44,45].

Finally, the use of new and existing fungicides is becoming more and more stringent. For example, depending on the commodity, the use of fungicides at postharvest is completely prohibited in some European countries or limited to a few registered chemicals. As detailed by [14], safety issues related to mycotoxins and foodborne pathogens also increase the need to find viable alternatives, such as products against postharvest citrus fruit pathogens.

In addition to the potency of 1,8-cineole as an antifungal compound, it is also a safe molecule that could be used directly on citrus fruit or its juice, as demonstrated by Bhandari and al. [46]. In addition, 1,8-cineole is a natural compound that occurs naturally in many citrus spp., has a good toxicology profile, and it has GRAS approval for use in food preparations [47].

## 4. Material and Methods

### 4.1. Chemicals and Growth Media

The major compound 1,8-cineole was obtained from the Mediterranean Flavour Society, Morocco. The chemical pesticides were obtained from Agripharma, Casablanca, Morocco: guazatine (KENOPEL20^®^), manufactured by SAOAS; orthophenyl phenol (DECCO OPP20^®^), manufactured by DECCO; imazalil (FUNGAFLOR 75 SP^®^), manufactured by JANSSEN PMP; thiabendazole (TECTO 500SC^®^), manufactured by SYNGENTA; and glyphosate (MAMBA TM DMA 480 SL^®^), manufactured by DowagroScience. Potato dextrose agar (PDA) and malt extract broth (ME) were obtained from Biokar, France. All media were autoclaved at 121 °C for 20 min and stored at 4 °C until use [48].

### 4.2. Sampling

Three small fragments of 1 cm^2^ from rotten mandarins were excised with the aid of sterile forceps, disinfected with 12% sodium hypochlorite solution for 10 min, and rinsed 3 times with sterile distilled water. The fragments were placed in 90 mm Petri dishes containing 20 mL of the PDA medium and then incubated at 27 °C for 5 days [41].

### 4.3. Morphological Identification of Fungal Species

The identification of isolates was based on morphological characteristics, growth speed, colony and reverse colony colour, and colony texture, dimensions, and pigmentation. The identification of isolates was performed according to the method of [38]. The isolates were stained with lactophenol (Merck, Darmstadt, Germany) and observed under an optical microscope (Optika, Ponteranica, Italy). For the absence or presence of partitions, the colour of the mycelial filaments, the mode of the ramification of the septums, and the differentiation of spores were analysed as previously described [42].

### 4.4. Spore Suspension

Spore suspensions of the isolates were generated based on the method described by [40]. Briefly, from 7-day-old Petri dishes, the spores were collected using a sterile spreading rod, after which the plate was flooded with 5 mL of 0.05% (*v*/*v*) Tween-20. The resulting spore suspension was adjusted using a counting chamber (surface area: 0.0025 mm^2^; depth: 0.2 mm) and diluted so that the resultant suspension was of the order of 10^6^ spores/mL.

### 4.5. In Vitro Antifungal Activity

The antifungal activity of the different substances was evaluated using the broth macro-dilution method [43]. This technique consists of incorporating the antifungal agent at a given concentration into the agar, maintained in liquid form at 42 °C. A series of Petri dishes with PDA medium containing different concentrations of the antifungal agents were prepared: OPP: 0.188, 0.375, 0.75, 1.5, and 3 mg/mL; IMZ: 0.05, 0. 1, 0.2, 0.4, and 0.8 mg/mL; TBZ: 0.375, 0.75, 1.5, 3, and 6 mg/mL; GUA: 0.125, 0.25, 0.5, 1, 2, and 4 mg/mL; GP: 0.06, 0.12, 0.24, 0.5, 1, 1.5, and 2 mg/mL; and 1,8-cineole: 0.025, 0.05, 0.1, 0.2, and 0.4 mg/mL. Plates were then inoculated with 10 µL of the spore suspension of 10^6^ spores/mL and incubated at 27 °C for 7 days. The diameter of the fungal colonies was measured daily during incubation. The percentage of mycelial growth inhibition (PI) was calculated according to the formula: PI (%) = (dt − Dt/dt) × 100, where dt and Dt represent the diameter in the absence and presence of the antifungal agent, respectively [3]. The MICs were recorded by reading the lowest chemical concentration that allowed no visible growth of the pathogen.

### 4.6. Spore Germination Assay

To test the effects of GP and the above-mentioned antifungal products on conidial germination, spore suspensions of *P. italicum, P. digitatum*, and *G. candidum* were prepared as described earlier. The germination medium was prepared by adding 2 mL of previously sterilised orange juice using a 0.2 µm filter to 98 mL of sterile distilled water [44]. Subsequently, in Eppendorf tubes, an aliquot of 400 μL was deposited. Next, 100 μL of the spore suspension was mixed with 500 μL of each concentration of different antifungals previously prepared in EM broth. In the negative control, we replaced the products with sterile distilled water, and 3 repetitions for each product were carried out. The tubes were incubated for 24 h at 27 °C under continuous stirring. The evaluation of spore germination inhibition was determined by spreading a volume of 100 μL from each tube on the surface of PDA Petri dishes of 90 mm [38]. After 7 days of incubation at 27 °C, the number of fungal colonies that appeared was counted [36].

### 4.7. The Effect of Glyphosate on Fungal Sensitivity to Different Antifungals In Vitro

The effect of different concentrations of GP on *P. digitatum, P. italicum,* and *G. candidum* development was examined in different concentrations of GP below and above the MIC value determined for the fungal pathogens: *P. digitatum*: 0.12, 0.17, 0.2, 0.24, 0.27, 0.3, and 0.5 mg/mL; for *P. italicum*: 0.12, 0.17, 0.2, 0.24, 0.27, 0.3, 0.4, 0.5, 0.8, and 1 mg/mL; and for *G. candidum*: 0.5, 0.6, 0.7, 1, 0.8, 1.2, 1.5, 1.6, 1.7, 1.8, and 2 mg/mL GP. Then, 50 µL of the spore suspension with a concentration of 10^6^ spores/mL was inoculated into tubes containing antifungal agents with lower MIC concentrations, and the inoculated tubes were incubated at 27 °C for 5 days. After incubation, successive subculturing was performed by transferring a volume of 50 µL from tubes where mycelia were present to tubes containing high concentrations of GP; this operation was carried out for all tubes until the new MIC was determined. The newly obtained resistant strains were reused again with the aim of determining the appropriate MIC against the postharvest antifungals and 1,8-cineole.

### 4.8. The Effect of Glyphosate on Fungal Sensitivity to Different Antifungals In Vitro in Contaminated Soil

The collected soil sample was sieved with a 2 mm mesh sieve and autoclaved twice at 121 °C for 30 min. Then, 100 g of the autoclaved soil was placed into pots and inoculated by dropping 10 mL of *P. italicum* and *P. digitatum* spore suspension [49]. The density of the inoculant was of the order of 10^8^ spores/mL. Afterwards, 1 mL of GP with a concentration of 0.12, 0.24, or 0.5 mg/mL was added to pots twice per week. Each week, a subsample of 1 g was taken from the pots and underwent a series of serial dilutions. A volume of 100 μL of diluted soil sample was inoculated into the PDA medium. After incubation for 7 days at 27 °C, the new MIC values were determined against the selected commonly used postharvest antifungals and 1,8-cineole.

### 4.9. Statistical Analysis

The results were expressed as the mean +/− standard deviation of three replicates. The significance of the results was verified using IBM SPSS Statistics Software (Armonk, NY, USA), Version 21, by testing the Analysis of Variance (ANOVA) using the Least Significant Difference (LSD) comparison test at a *p* value less than 0.05.

## 5. Conclusions

1,8-Cineole has shown interesting antifungal activity without inducing microbial tolerance to it. It can be suggested that 1,8-cineole could be further developed, especially for the control of many fungal species responsible for different phytopathogenic forms; it also remains an effective alternative to chemical antifungals. This is in the interest of consumer health and the environment. Moreover, the risk of transferring resistant genes to humans should not be neglected. As the results of this study on fungal tolerance to glyphosate and other fungicides are very new, these phenomena need conclusive experimental support; the influence of pH on the potency and solubility of fungicides and on fungal growth (any assay should include pH as a controlled or reported factor) should be investigated, and more conventional methods of quantifying fungicide resistance are needed. On the other hand, empirical experiments showing the ability of 1,8-cineole to control fruit diseases caused by these pathogens are needed, preferably under natural infection conditions, including an evaluation of the risk of phytotoxicity to the fruit. In addition, future studies are recommended to confirm these results at the genetic level in order to fully understand how resistance and cross-resistance develop in crops against chemical pesticides because, for this urgent problem, alternative solutions should not be overlooked.

## Figures and Tables

**Figure 1 molecules-27-08300-f001:**
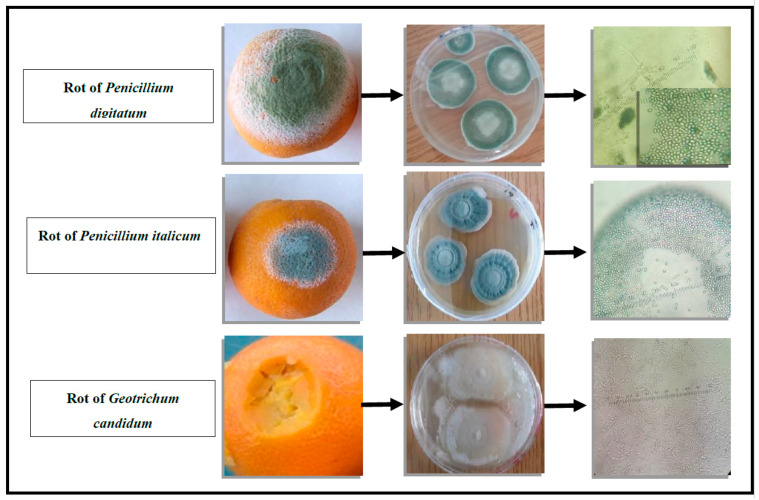
The macroscopic and microscopic aspects of *P. digitatum*, *P. italicum*, and *G. candidum*.

**Figure 2 molecules-27-08300-f002:**
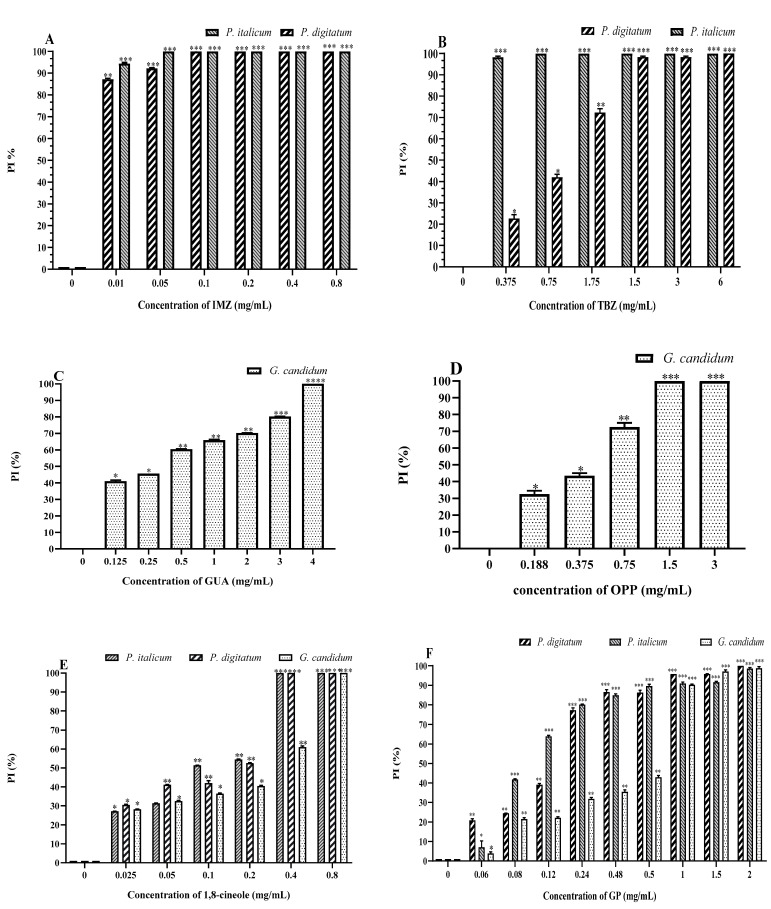
The percentage of fungal inhibition (PI) of IMZ (**A**), TBZ (**B**), GUA (**C**), OPP (**D**), 1.8-cineole (**E**), and GP (**F**). *p* < 0.05 (*); *p* < 0.01 (**); *p* < 0.001 (***); *p* < 0.0001 (****).

**Figure 3 molecules-27-08300-f003:**
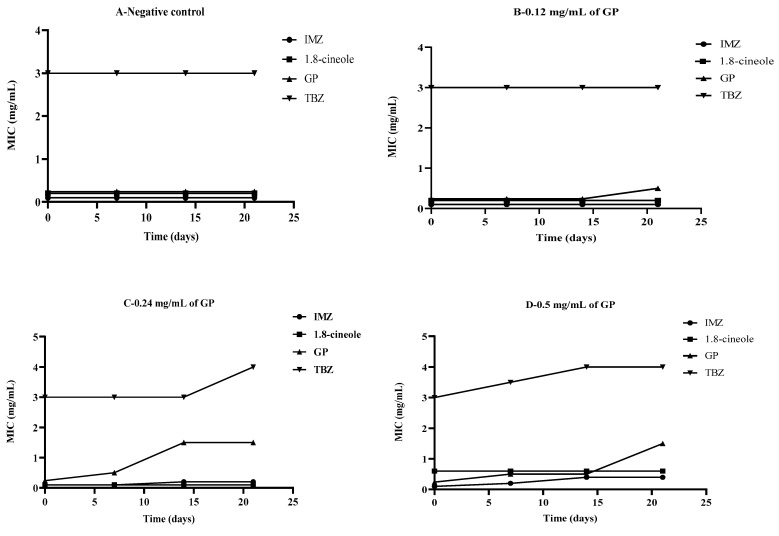
Development of *P. digitatum* tolerance to antifungal treatment after soil treatments with GP. (**A**), negative control; (**B**), 0.12 mg/mL of GP; (**C**), 0.24 mg/mL of GP; (**D**), 0.5 mg/mL of GP.

**Figure 4 molecules-27-08300-f004:**
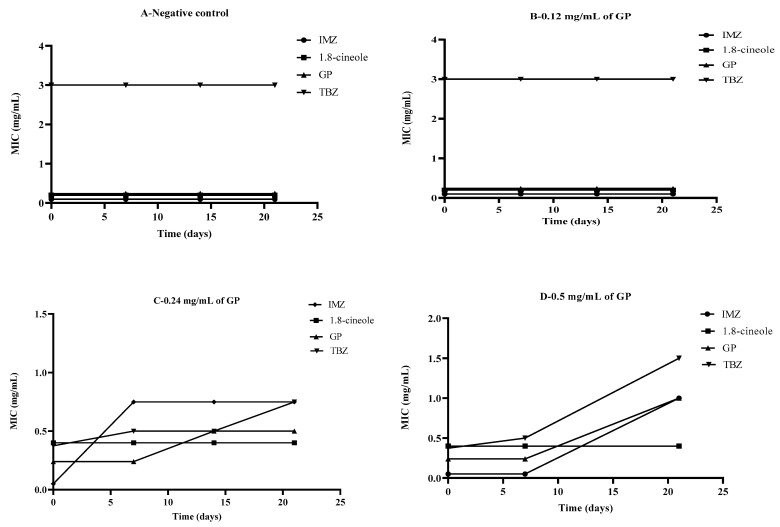
Development of *P. italicum* tolerance to antifungal treatments after soil treatment with GP. (**A**), negative control; (**B**), 0.12 mg/mL of GP; (**C**), 0.24 mg/mL of GP; (**D**), 0.5 mg/mL of GP.

**Table 1 molecules-27-08300-t001:** Minimum inhibitory concentrations of IMZ, TBZ, OPP, 1,8-cineole, and GP against citrus fruit fungal pathogens.

Antifungal Agent	Fungal Pathogen	MIC (mg/mL)
IMZ	*P. digitatum*	0.01
	*P. italicum*	0.05
TBZ	*P. digitatum*	3
	*P. italicum*	0.375
GUA	*G. candidum*	4
OPP	*G. candidum*	1.5
1,8-Cineole	*P. digitatum*	0.4
	*P. italicum*	0.2
	*G. candidum*	0.4
GP	*P. digitatum*	0.24
	*P. italicum*	0.24
	*G. candidum*	1

**Table 2 molecules-27-08300-t002:** Effects of the antifungal agents on spore germination of *P. italicum* (*n* = 3).

Antifungal Agent	Number of Colonies (CFU)				
IMZ	0.05 mg/mL	0.1 mg/mL	0.2 mg/mL	0.4 mg/mL	0.8 mg/mL
	9 ± 20	0	0	0	0
TBZ	0.375 mg/mL	0.75 mg/mL	1.5 mg/mL	3 mg/mL	6 mg/mL
	11 ± 30	0	0	0	0
1,8-Cineole	0.025 mg/mL	0.05 mg/mL	0.1 mg/mL	0.2 mg/mL	0.4 mg/mL
	245 ± 15	112 ± 10	59 ± 66	3 ± 10	0
GP	0.12 mg/mL	0.24 mg/mL	0.5 mg/mL	1 mg/mL	1.5 mg/mL
	19 ± 20	4 ± 10	0	0	0
Negative control	≥300	≥300	≥300	≥300	≥300

**Table 3 molecules-27-08300-t003:** Effects of the antifungal agents on spore germination of *P. digitatum* (*n* = 3).

Antifungal Agent	Number of Colonies (CFU)				
IMZ	0.05 mg/mL	0.1 mg/mL	0.2 mg/mL	0.4 mg/mL	0.8 mg/mL
	11 ± 20	0	0	0	0
TBZ	0.375 mg/mL	0.75 mg/mL	1.5 mg/mL	3 mg/mL	6 mg/mL
	90 ± 50	25 ± 30	0	0	0
1,8-Cineole	0.025 mg/mL	0.05 mg/mL	0.1 mg/mL	0.2 mg/mL	0.4 mg/mL
	270 ± 11	188 ± 14	47 ± 90	4 ± 50	0
GP	0.12 mg/mL	0.24 mg/mL	0.5 mg/mL	1 mg/mL	1.5 mg/mL
	56 ± 30	9 ± 20	0	0	0
Negative control	≥300	≥300	≥300	≥300	≥300

**Table 4 molecules-27-08300-t004:** Effects of the antifungal agents on spore germination of *G. candidum* (*n* = 3).

Antifungal Agent	Number of Colonies (CFU)				
OPP	0.188 mg/mL	0.375 mg/mL	0.75 mg/mL	1.5 mg/mL	3 mg/mL
	≥300	88 ± 90	48 ± 40	0	0
GUA	0.125 mg/mL	0.05 mg/mL	1 mg/mL	2 mg/mL	4 mg/mL
	≥300	78 ± 90	16 ± 50	4 ± 20	0
1,8-Cineole	0.025 mg/mL	0.05 mg/mL	0.1 mg/mL	0.2 mg/mL	0.4 mg/mL
	290 ± 15	90 ± 11	42 ± 50	0	0
GP	0.12 mg/mL	0.24 mg/mL	0.5 mg/mL	1 mg/mL	1.5 mg/mL
	150 ± 18	52 ± 20	3 ± 10	0	0
Negative control	≥300	≥300	≥300	≥300	≥300

**Table 5 molecules-27-08300-t005:** In vitro evaluation of fungal development of tolerance to GP.

Fungal Pathogen	Initial MIC Value	MIC after MSubculturing
*P. digitatum*	0.24 mg/mL	1 mg/mL
*P. italicum*	0.24 mg/mL	0.5 mg/mL
*G. candidum*	1 mg/mL	2 mg/mL

**Table 6 molecules-27-08300-t006:** Effect of GP on fungal tolerance to IMZ, TBZ, GUA, OPP, and * 1,8-cineole in vitro.

Fungal Pathogen	MIC of IMZ before and after Subculturing	MIC of TBZ before and after Subculturing	MIC of GUA before and after Subculturing	MIC of OPP before and after Subculturing
*P. digitatum* (7 subcultures)	from 0.1 to 0.4 mg/Ml	No change (1.5 mg/mL)	-	-
*P. italicum* (10 subcultures)	from 0.5 to 0.8 mg/Ml	from 0.375 to 3 mg/mL	-	-
*G. candidum* (12 subcultures)	-	-	from 1.5 to 3 mg/mL	from 1 to 2 mg/mL

* No change in MIC was detected for 1,8-cineole.

## Data Availability

All data included in the main text.

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
