# Peer review of "Treatment with Glyphosate Induces Tolerance of Citrus Pathogens to Glyphosate and Fungicides but Not to 1,8-Cineole"

_molecules, 2022, doi:10.3390/molecules27238300_

Round 1

Reviewer 1 Report

Rev molecules-1983706 Oct 14 2022

Subjects in this work include evaluation of the toxicity and influence of glyphosate on the growth and development of fungicide resistance by several citrus fruit postharvest pathogens, with the inclusion of 1,8-cineole, is also known as eucalyptol, in these tests. Issues with this manuscript:

1.     A critical variable, pH, was not controlled or reported or mentioned, and it can greatly influence the activity of the many of the compounds tested. 

2.     Fungicides and glyphosate used in these tests were commercial agricultural formulations, with additives that could influence the results. Authentic compounds with high purity should be used. The author reports a very novel result, both glyphosate inhibition and resistance of several fungi, and these phenomena need conclusive experimental support.

3.     Many minor revisions are needed in word selection, excessively long sentences, passive voice, word choices, significant figures, etc. 

4.     Unconventional presentation of dose – mortality results. There are simple ways to describe the magnitude of inhibition in a quantitative fashion. It is conventional to apply probit or logit analysis to inhibition data, with results expressed as ED50 or LD50 concentrations where 50% inhibition occurred, often with 95% confidence intervals, and this is easy for the reader to understand. Some of the data presented could be analyzed this way. Another approach would be resistance frequency in the spore mortality assays – a fictional example: The control had 1/250,000 = one spore became resistant of 250,000 exposed, when GP was added, the frequency was 1/10,000. 

The author states “…1,8-cineole may be recommended as an alternative to conventional fungicides.”  To support this statement, empirical experiments showing control of the diseases on fruit caused by these pathogens by 1,8-cineole are needed, preferably under natural infection conditions, including an evaluation of the risk of phytotoxicity to the fruit.  If effective, this would be a major contribution. Recent by practical postharvest work in this subject by Bhandari et al (2021 J. Hortic. Sci 16:77-90) and others should be cited.  It (1,8-cineole) has a good toxicology profile, already occurs naturally in many citrus spp., and it has GRAS approval for many applications in foods, although its disposal raises ecotoxicity concerns (Bhowal & Gopal, 2016. Eucalyptol: Safety and Pharmacological Profile. J Pharm Sci DOI: 10.5530/rjps.2015.4.2and Salvatore, et al, 2022. Essential oils in citrus fruit ripening and postharvest quality Horticulturae 8:396).  

The result that glyphosate had substantial antifungal activity is novel and needs more experimental support. The glyphosate used was a commercial formulation that may have had antifungal additives in it that confounded interpretation of its effect on the fungicides. Mesnage et al (2022 Toxicological Sciences, Volume 186, Issue 1, March 2022, Pages 83-101,  https://doi.org/10.1093/toxsci/kfab143) recently reported that formulated glyphosate was more cytotoxic to mammalian cells than glyphosate alone.

The pH of the solutions tested is not reported and must be shown. The typical glyphosate formulation has a low pH and pH greatly influences the potency and solubility of fungicides. It may have varied among the test concentrations of fungicides, when the glyphosate was added, or even pH alone may have directly inhibited some of these fungi (especially P. digitatum). Glyphosate has four ionizable groups, each with its own pKa, and the state of each of these could influence its activity. The toxicity of these fungicides is greatly influenced by pH, particularly those that ionize, such as imazalil and sodium orthophenylphenol. For example, from Smilanick et al. (2005. Plant Dis. 89:640-648): “Estimated concentrations of imazalil (IMZ) in potato-dextrose broth–Tris that caused 50% reduction in the germination of spores (ED50) of an IMZ-sensitive isolate M6R at pH 4, 5, 6, and 7 were 0.16, 0.11, 0.015, and 0.006 μg/ml, respectively. ED50 IMZ concentrations of an IMZ-resistant isolate D201 at pH 4, 5, 6, and 7 were 5.9, 1.4, 0.26, and 0.07 μg/ml, respectively.” In general, the neutral forms are much more potent than the ionized forms. For example, at the pH of most glyphosate formulations (4-5), imazalil antifungal activity is much lower than at higher pH. In Table 2, IMZ was greatly inhibitory beginning at 0.05 mg/ml and above. This concentration equals 50 mg/L (=50 ppm). At pH 7, the IMZ MIC should be less than 0.1 mg/L. P. digitatum is intolerant of high pH alone, and its germination is inhibited at pH 8 and above.

The exposure of Penicillium spp. and Geotrichum isolates to glyphosate and these fungicides together under commercial citrus cultural practices seems unlikely to occur, and glyphosate residues on the fruit would be vanishingly small.  Weed control applications are typically early season, long before harvest, and the trees are never the target of the herbicide applications. Glyphosate kills citrus trees. Citrus are not like GMO glyphosate-resistant crops, where glyphosate is directly applied to the crop and a glyphosate residue are routinely present on them.  Glyphosate residues in soil decline, sometimes very rapidly. Geotrichum citri-aurantii survives and persists in citrus grove soils, in contrast Penicillium digitum and P. italicum survive primarily on fallen fruit and do not colonize soil.

The fungicides characterized in this manuscript are all exclusively applied after harvest. Selection pressure to favor the proliferation of fungicide resistant isolates begins when they are applied in packinghouses, where glyphosate is not used and its residues are likely very rare.

Specific comments

L9-10 The generic names of these fungicides should not be capitalized.

L9 The citrus infecting Geotrichum that causes citrus sour rot is Geotrichum citri-aurantii (imperfect) or Galactomyces citri-aurantii (perfect). See: McKay, et al. 2012. Distinguishing Galactomyces citri-aurantii from G. geotrichum and characterizing population structure of the two postharvest sour rot pathogens of fruit crops in California. Phytopathology 102:528-38.  doi: 10.1094/PHYTO-05-11-0156.

L13 The term “cross-resistance” is not used correctly here. It means resistance to one compound is accompanied by resistance to a second compound. 

L13-18 Exceptionally long, hard to follow sentence here. I suggest: “IMZ most effectively inhibited mycelial growth and spore germination of P. digitatum and P. italicum, with minimum inhibitory concentrations (MIC) of 0.01 and 0.05 mg/ml, respectively, followed by 1,8-cineole, GP, and TBZ. 1,8-Cineole and GP more effectively inhibited mycelial growth and spore germination of G. citri-aurantii, with minimum inhibitory concentrations (MIC) of 0.2 and 1.0 mg/ml, respectively, than OPP or GUA.

L21 The terms “tolerance” and “resistance” are used interchangeably throughout the manuscript and it is confusing. Common definitions in use of these words:  “Tolerance” is an activation of existing detoxification mechanisms to evade mortality, and is of a lower order than “resistance” and not inheritable. “Resistance” is affected by a heritable mechanism and provides a higher magnitude of protection. 

L128-130. Concentration series to determine levels of fungicide resistance are described here and mycelial growth is described by MIC. This is acceptable, but a more conventional way of quantify fungicide resistance, is to express inhibition as the ED50 concentration of both spore germination and mycelial growth. 

L52 The citation 8 by Chen et al. 2019 does not support this statement that these fungi are mycotoxin producers. Chen et al. describes inhibition of P. italicum by 7-demethoxytylophorine and not mycotoxin production of this or other fungi. Reviews describing these fungi indicate mycotoxins are not a concern (Pitt & Hocking 2009 “Fungi and Food Spoilage” Springer)

L158-162 The Penicillia will germinate well in aqueous solutions, but they are not aquatic fungi and grow poorly when immersed in aqueous solutions for more than a day or so.  To be re-cultured in a liquid medium, particularly still culture that would rapidly become anerobic, is odd for these strictly aerobic fungi. 

Author Response

We thank you for reviewing our manuscript and suggesting valuable points for improvement in the current work. We have carefully considered all your questions and comments and highlighted each change made in the revised manuscript (with the colour yellow). The point-by-point responses to your comments and questions are attached here.

Reviewer 2 Report

Peer Review Report

Peer review report 3 on “Treatment with glyphosate induces tolerance of citrus patho- 2 gens to glyphosate and fungicides but not to 1,8-cineole”

1.     Original Submission

1.1  Recommendation

Major/minor revision

2.     Comments to Author:

Ms. Ref. No.: molecules-1983706

Title: Treatment with glyphosate induces tolerance of citrus patho- 2

gens to glyphosate and fungicides but not to 1,8-cineole

Overview and general recommendation:

Citrus fruits are exposed to several citrus pathogens such as Penicillium italicum, Penicillium digitatum, and Geotrichum candidum, causing serious yield losses worldwide.

Pesticides are commonly used as antifungals.
Alternatives to chemical fungicides such as studies of plant disease resistance has gained much attention. In this study, the sensitivity of fungal pathogens of citrus fruit to these pesticides and 1,8-cineole was investigated and the effect of Glyphosate (GP) on the development of cross-resistance to other chemicals was monitored over a period of three weeks.

The author described that Imazalil (IMZ) was the most effective fungicide against P. digitatum and P. italicum, followed by followed by 1,8-cineole and GP and then Thiabendazole (TBZ). In addition, 1,8-cineole and GP were very effective against G. candidum compared to other fungicides.

I came away with some questions to be able to recommend this paper for publication as it stands. Therefore, I truly recommend that some of the description (discussion) to be fit in the introduction not in the discussion section. I explain my concerns in more detail below. I ask that the authors specifically address each of my comments in their response.

2.1  Major comments:

1.     One of the concerns I have about the paper is with respect to the Introduction (Page 2. Line 47). Is this term correct? “pseudo-fungal pathogens” – what does it mean?

2.     Session 3.4: “The ability of P. digitatum, P. italicum and G. candidum to develop resistance against GP was examined in vitro by sub-culturing the fungal strains in culture medium containing increasing concentration of GP” – The MIC values described in Table 5 were observed after 3 weeks? This was not clear for me. 

3.     (Page 2. Line 53): “Indeed, these rots cause significant economic losses [9], [10].” This sentence is repetitive. The significant economic losses were described in the paragraphs before.

4.     Ref [1]: It is missing the accessed date).

5.     (Page 4. Line 183, Line 200, Line 221): “Penicillium and Geotrichum” (genera should be in italicum. Please check italicum format in all the manuscript).

6.     Line 216: What is “PI”? Please describe definition.

7.     Figure 1. “Rot of Penicillium digitatum” is out of the box

Anonymous

Author Response

We thank you for reviewing our manuscript and suggesting valuable points for improvement in the current work. We have carefully considered all your questions and comments and highlighted each change made in the revised manuscript (with the colour green). The point-by-point responses to your comments and questions are attached here.

Round 2

Reviewer 1 Report

I recommend the revised manuscript for publication, but I find the lack of attention to pH the most significant weakness and the added statements by the authors should mention more about this. The results do not conclusively support the glyphosate-fungicide interaction they describe, but the idea is very original and merits publication for this reason. For example, the magnitude of pH influence on imazalil and SOPP performance is very large - orders of magnitude. P. digitatum growth even ceases at pH 8 and above, so any assay with it should include pH as a controlled or reported factor. The addition of recent citations of mycotoxin production improve the manuscript. The 1,8-cineole should only be called promising after a commercial simulation trial has been done with natural infection and where phytotoxicity risk has been examined. My associates who evaluate chemical and biological products for commercial postharvest use have told me many treatments that are called promising by their authors in the literature have applied this term prematurely, and the term promising should only be applied after effectiveness to control natural infections and determine the risk of phyotoxicity are known. This empirical, practical information should be shown in the initial publication describing a treatment.

Author Response

Dear respected Reviewer

Firstly, I would like to thank you very much for the important and constructive suggestions given for our manuscript, which will certainly improve its scientific value.

Secondly, as you know, we tested commercial products, which are commonly used directly by farmers without any modification. In addition, we tried to find a balance between internal and external validity in order to show that the causal relationship we were testing is not influenced by other factors or variables, and at the same time to show that the results of the study can be generalized to other contexts. On the other hand, prior to taking into account your interesting comments, we have added your suggestion as a limitation (highlighted in the manuscript) of the study. Finally, the term “promising” that we were using has been removed.

We hope that the revised version has satisfactorily been improved and is now suitable for publication in Molecules.

Kind regards